DATA RELEASE

# A chromosome-scale draft genome sequence of horsegram (*Macrotyloma uniflorum*)

Kenta Shirasawa[1], Rakesh Chahota[2], Hideki Hirakawa[1], Soichiro Nagano[1,4], Hideki Nagasaki[1], Tilak Sharma[3] and Sachiko Isobe[1,*]

1 Kazusa DNA Research Institute, 2-6-7 Kazusa-kamatari, Kisarazu, Chiba 292-0818, Japan
2 Department of Agricultural Biotechnology, CSK Himachal Pradesh Agricultural University, Palampur, Himachal Pradesh 176062, India
3 ICAR – Indian Institute of Agricultural Biotechnology, Ranchi, Jharkhand 834010, India
4 Forest Tree Breeding Center, Forestry and Forest Products Research Institute, 3809-1 Ishi, Juo, Hitachi, Ibaraki 319-1301, Japan

## ABSTRACT

Horsegram (*Macrotyloma uniflorum* [Lam.] Verdc.) is an underutilized warm-season diploid legume ($2n$ = 20, 22). Because of its ability to grow under water-deficient and marginal soil conditions, horsegram is a preferred choice in the era of global climate change. In recognition of its potential as a crop species, we generated and analyzed a draft genome sequence for a horsegram variety, HPK-4. Ten chromosome-scale pseudomolecules were created by aligning Illumina scaffold sequences onto a linkage map. The total length of the ten pseudomolecules was 259.2 Mbp, covering 89% of the total length of the assembled sequences. A total of 36,105 genes were predicted on the assembled sequences. Diversity analysis of 89 horsegram accessions by dd-RAD-Seq identified 277 single nucleotide polymorphisms (SNPs), suggesting narrow genetic diversity among the horsegram accessions. This is the first attempt to generate a draft genome sequence of horsegram and will provide a reference for sequence-based analysis of horsegram germplasm.

**Subjects** Animal and Plant Sciences, Plant Genetics, Molecular Genetics

**Submitted:** 18 June 2021

\* Corresponding author. E-mail: sisobe@kazusa.or.jp

Preprint submitted at https://doi.org/10.1101/2021.01.18.427074

# DATA DESCRIPTION
## Background

Horsegram (*Macrotyloma uniflorum* [Lam.] Verdc.) (NCBI:txid271171), is an underutilized warm-season diploid legume ($2n$ = 20, 22). It belongs to the Fabaceae family of the Phaseoleae tribe, and is cultivated mainly in semi-arid regions of the world. On the Indian subcontinent, horsegram is consumed primarily as a food legume, whereas in Africa and Australia it is grown mainly for use as a concentrated animal feed and fodder. This self-pollinating plant is thought to have originated in Africa because most of its 32 wild species exist there [1], and the Northwestern Himalayan region is considered its secondary center of origin [2]. Horsegram may have been domesticated as *M. uniflorum* var. uniflorum in the southern part of India, but its probable progenitor, *M. axillare*, has not been reported in India. Therefore, the process by which cultivated horsegram was domesticated from its wild ancestors has not yet been established [3].

Because of its ability to grow under water-deficient and marginal soil conditions, horsegram is a preferred choice in the era of global climate change. Horsegram contains

16.0–30.4% protein [4], and constitutes an important source of dietary protein for the undernourished population in south Asia. In addition, the seeds are a rich source of lysine and vitamins [5], and its antioxidant, antimicrobial, and unique antilithiatic properties make it a food of nutraceutical importance [6–8]. As a result of horsegram's medicinal importance and ability to thrive under drought-like conditions, the US National Academy of Sciences has identified this legume as a potential food source for the future [9].

## Context

The existence of many wild and unsolicited characteristics makes horsegram a less favorable legume for commercial cultivation, although it does possess numerous attributes that make it a potential food legume for warm arid regions. In addition, there is a lack of genetic and molecular tools with which to genetically enhance horsegram. To elucidate the potential of this food legume species, we generated and analyzed a draft genome sequence for HPK-4, a horsegram cultivar commercially released by CSK Himachal Pradesh Agricultural University (HPAU), Palampur, India. This variety, which has dark-brown seeds, is under cultivation in many parts of the Northwestern Indian Himalayan region. It is resistant to anthracnose (*Colletotrichum truncatum*) and tolerant to abiotic stresses such as drought, salinity, and heavy metals. This is the first attempt to generate a draft genome sequence of this 'orphan', but it is an important food legume species and will provide a reference for sequence-based analysis of horsegram germplasm to elucidate the genetic bases of important traits.

## METHODS

### Whole genome sequencing and assembly of horsegram

The genome sequences of a horsegram variety, HPK-4, bred at CSK-HPAU, were generated from a paired-end (PE) library by Illumina HiSeq 2000 with a total length of 37.9 Gbp (gigabase pairs) [10]. All data analysis for this study was performed on Linux servers running Red Hat Enterprise Linux Server 7.1. Using the Jellyfish v1.1.6 program (RRID: SCR_005491) [11], the genome size of HPK-4 was estimated to be approximately 343.6 Mbp (megabase pairs) (Figure 1). The parameters used in the analysis are listed in Table 1.

The Illumina PE reads were assembled by SOAPdenovo2 r223 (RRID: SCR_014986) [12] with *k*-mers of 61 and 81, and contigs were generated with total lengths of 352.2 Mbp (*k*-mer = 81) and 389.3 Mbp (*k*-mer = 61) (Table 2). The contigs constructed with *k*-mer = 81 were selected and scaffolded with mate-pair (MP) reads with insert sizes of 2, 5, 10, and 15 Kbp (kilobase pairs) by using SSPACE v2.0 (RRID: SCR_005056) [13]. The number of generated scaffolds was 6227 after gap filling by GapFiller [14] and excluding contamination. The total length of the scaffolds was 297.1 Mbp (Assembly 1, Table 2), which was approximately 55–92 Mbp shorter in length than the estimated genome size of HPK-4.

We speculated that the shorter observed length of the total scaffolds may have been caused by misintegration of repeat sequences by SSPACE v2.0. Therefore, we performed subsequent assemblies using two programs, Platanus v1.2.1 (RRID: SCR_015531) [15] and MaSuRCA v2.3.2 (RRID: SCR_010691) [16]. The total ATGC lengths of the scaffolds were not significantly different among the three assemblies: 284.4 Mbp in SOAPdenovo-SSPACE (Assembly 1, before excluding contamination; Table 2), 277.0 Mbp in Platanus (Assembly 2), and 299.1 Mbp in MaSuRCA (Assembly 3).

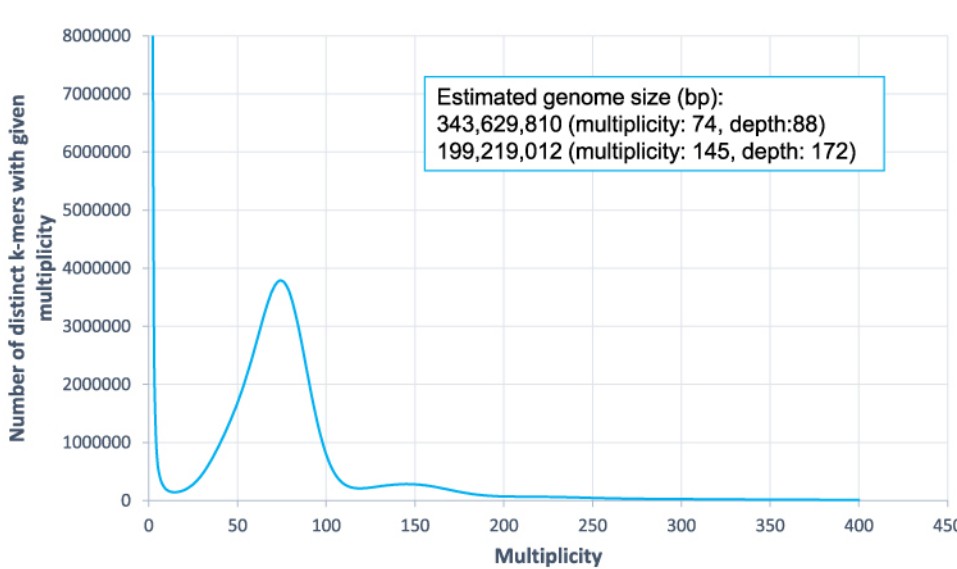

**Figure 1.** Genome size estimation using Jellyfish with the distribution of the number of distinct *k*-mers (*k* = 17) with the given multiplicity values.

**Table 1.** Parameters used in each program.

| Program name | Parameters or BUSCO data set | Comments |
|---|---|---|
| Jellyfish v1.1.6 | –m 17 –s 1000000000 –t 32 –C | |
| SOAPdenovo2 r223 | –K (61 71 81 91) –R –F –p 8 | |
| SSPACE v2.0 | –x 0 –z 0 –k 3 –a 0.7 –n 15 –T 8 –g 0 –v 1 | |
| GapFiller v1.10 | –m 30 –o 5 –r 0.7 –n 10 –d 50 –t 10 –g 0 –T 8 –i 1 | |
| Platanus v1.2.1 | –t 12 –m 300 | |
| MaSuRCA v2.3.2 | Default Parameters | |
| TruSPAdes v3.6.2 | Default Parameters | |
| RepeatMasker v3.2.9 | –poly –x –lib | |
| RepeatScout v1.0.5 | Default Parameters | |
| BRAKER1 v1.9 | Default Parameters | |
| BUSCO v3.0 | Embryophyta, odb10 | |
| Samtools 0.1.19 | samtools mpileup mpileup –d 10000000 –D –u | |
| bcftools 0.1.19 | bcftools view –c –g –v | |
| vctools 0.1.12b | vcftools_0.1.12b/bin/vcftools –remove-indels –min-alleles 2 –max-alleles 2 –minDP 5 –minQ 214 –max-missing 1 | SNP filterfing for 8 $F_2$ WGS |
| vctools 0.1.12b | vcftools_0.1.12b/bin/vcftools –remove-indels –min-alleles 2 –max-alleles 2 –minDP 10 –minQ 50 –max-missing 0.2 | SNP filterfing for 214 $F_2$TAS |
| vctools 0.1.12b | vcftools_0.1.12b/bin/vcftools –remove-indels –min-alleles 2 –max-alleles 2 –minDP 5 –minQ 999 –max-missing 0.5 –maf 0.05 | SNP filtering for 89 population |
| JoinMap 4 | Kosambi's mapping function, linkage with rec. frec. Smaller than 0.4 and a LOD lather than 1.0, Goodness-of-fit for removal of loci = 5.0, Number of added loci after which to perform a ripple = 1, Third round = yes | |

Meanwhile, an Illumina synthetic long-reads (SLR) library was constructed with high-molecular-weight cellular DNA using a TruSeq synthetic long-read DNA library prep kit (Illumina). Sequences were generated by Illumina HiSeq 2000 and MiSeq systems with read lengths of 93 nt and 251 nt, respectively. The SLR reads were synthesized through the



**Table 2.** Statistics of de novo whole genome assembly.

| File name | SOAPdenovo Contigs | | Assembly 1 SOAPdenovo/SSPACE/GapFiller (*k*-mer = 81) | | Assembly 2 Platanus | Assembly 3 MaSuRCA | Assembly 4 TruSPAdes | Assembly 5 GMcloser | MUN_r1.1 | MUN_r1.11 | MUN_r1.11 pseudomolecule |
|---|---|---|---|---|---|---|---|---|---|---|---|
| Input reads | PE | PE | PE + MP | PE + MP | PE + MP | PE + MP | SLR | Assembly 1 + Assembly 4 | Assembly 5 | MUN_r1.1 | MUN_r1.11 + Linkage map |
| Comments | *k*-mer = 61 | *k*-mer = 81 | Include contaminat | Exclude contaminat | Include contaminat | Include contaminat | | Exclude contaminat | ≥500bp | Scaffolds revised | |
| **All** | | | | | | | | | | | |
| Number of sequences | 1,534,576 | 779,101 | 7,123 | 6,228 | 62,323 | 17,400 | 374,253 | 6,228 | 3,495 | 3,497 | 10 |
| Total length (bp) | 389,388,347 | 352,263,669 | 297,816,217 | 297,127,168 | 287,695,252 | 313,146,882 | 1,357,659,30 | 295,740,202 | 294,688,765 | 294,688,765 | 259,245,825 |
| Average length (bp) | 254 | 452 | 41,811 | 47,708 | 4,616 | 17,997 | 3,628 | 47,486 | 84,317 | 84,269 | 25,924,583 |
| Max length (bp) | 43,864 | 86,786 | 13,495,995 | 13,495,995 | 13,114,378 | 9,397,721 | 79,948 | 13,482,853 | 9,844,273 | 9,844,273 | 33,386,276 |
| Min length (bp) | 62 | 82 | 300 | 300 | 100 | 71 | 1,500 | 146 | 500 | 500 | 15,505,026 |
| N50 length (bp) | 2,602 | 6,108 | 3,571,813 | 3,571,813 | 4,221,442 | 2,147,735 | 4,120 | 3,568,883 | 2,818,555 | 2,818,555 | 28,154,654 |
| A | 137,634,352 | 123,511,855 | 98,936,270 | 98,713,635 | 95,878,438 | 103,574,375 | 440,896,819 | 100,065,275 | 99,718,915 | 99,718,915 | 88,615,763 |
| T | 128,359,494 | 116,788,629 | 98,000,497 | 97,829,343 | 96,128,274 | 103,392,204 | 440,199,341 | 99,104,678 | 98,784,555 | 98,784,555 | 88,538,203 |
| G | 61,810,612 | 56,330,394 | 43,804,355 | 43,679,038 | 42,484,816 | 46,002,912 | 238,337,484 | 44,442,744 | 44,254,557 | 44,254,557 | 38,863,122 |
| C | 61,583,889 | 55,632,791 | 43,617,622 | 43,524,196 | 42,490,698 | 46,133,757 | 238,219,308 | 44,268,956 | 44,083,227 | 44,083,227 | 38,986,862 |
| N | 0 | 0 | 13,457,473 | 13,380,956 | 10,713,026 | 14,043,634 | 6,350 | 7,858,549 | 7,847,511 | 7,847,511 | 4,241,875 |
| Total (ATGC, bp) | 389,388,347 | 352,263,669 | 284,358,744 | 283,746,212 | 276,982,226 | 299,103,248 | 1,357,652,95 | 287,881,653 | 286,841,254 | 286,841,254 | 255,003,950 |
| GC% (ATGC) | 31.7 | 0 | 30.7 | 30.7 | 30.7 | 30.8 | 35.1 | 30.8 | 30.8 | 30.8 | 30.5 |
| **≥300 bp** | | | | | | | | | | | |
| Number of sequences | 96,834 | 85,229 | 7,123 | 6,228 | 13,045 | 17,107 | 374,253 | 6,226 | - | - | - |
| Total length (bases) | 255,385,256 | 270,010,759 | 297,816,217 | 297,127,168 | 281,104,166 | 313,093,559 | 1,357,659,30 | 295,739,758 | - | - | - |
| Average length (bases) | 2,637 | 3,168 | 41,811 | 47,708 | 21,549 | 18,302 | 3,628 | 47,501 | - | - | - |
| **≥500 bp** | | | | | | | | | | | |
| Number of sequences | 72,097 | 56,065 | 3,945 | 3,468 | 8,514 | 13,654 | 374,253 | 3,469 | 3,495 | 3,497 | |
| Total length (bases) | 245,981,429 | 258,994,765 | 296,598,533 | 296,074,149 | 279,298,951 | 311,725,703 | 1,357,659,30 | 294,688,765 | 294,688,765 | 294,688,765 | |
| Average length (bases) | 3,412 | 4,619 | 75,183 | 85,373 | 32,805 | 22,830 | 3,628 | 84,949 | 84,317 | 84,269 | |
| **≥1 Kbp** | | | | | | | | | | | |
| Number of sequences | 52,710 | 39,176 | 1,976 | 1,842 | 2,716 | 9,787 | 374,253 | 1,836 | 1,862 | 1,864 | |
| Total length (bases) | 232,331,716 | 247,369,100 | 295,266,774 | 294,976,239 | 275,198,779 | 308,936,166 | 1,357,659,30 | 293,585,247 | 293,585,247 | 293,585,247 | |
| Average length (bases) | 4,408 | 6,314 | 149,427 | 160,139 | 101,325 | 31,566 | 3,628 | 159,905 | 157,672 | 157,503 | |
| **≥2 Kbp** | | | | | | | | | | | |
| Number of sequences | 27,007 | 23,688 | 1,205 | 1,186 | 511 | 4,079 | 190,853 | 1,176 | 1,202 | 1,204 | |
| Total length (bases) | 185,671,228 | 219,707,018 | 294,019,523 | 293,893,601 | 272,047,316 | 299,058,101 | 960,490,176 | 292,496,469 | 292,496,469 | 292,496,469 | |
| Average length (bases) | 6,875 | 9,275 | 244,000 | 247,802 | 532,382 | 73,317 | 5,033 | 248,721 | 243,341 | 242,937 | |



**Table 2.** (Continued)

| File name | SOAPdenovo Contigs | | Assembly 1 SOAPdenovo/SSPACE/ GapFiller (*k*-mer = 81) | | Assembly 2 Platanus | Assembly 3 MaSuRCA | Assembly 4 TruSPAdes | Assembly 5 GMcloser | MUN_r1.1 | MUN_r1.11 | MUN_r1.11 pseudomolecule |
|---|---|---|---|---|---|---|---|---|---|---|---|
| Input reads | PE | PE | PE + MP | PE + MP | PE + MP | PE + MP | SLR | Assembly 1 + Assembly 4 | Assembly 5 | MUN_r1.1 | MUN_r1.11 + Linkage map |
| Comments | *k*-mer = 61 | *k*-mer = 81 | Include contaminat | Exclude contaminat | Include contaminat | Include contaminat | | Exclude contaminat | ≥500bp | Scaffolds revised | |
| ≥3 Kbp | | | | | | | | | | | |
| Number of sequences | 15,668 | 16,505 | 1,084 | 1,073 | 395 | 3,261 | 70,396 | 1,056 | 1,082 | 1,084 | |
| Total length (bases) | 141,491,218 | 191,540,893 | 293,553,905 | 293,455,161 | 271,611,649 | 295,920,059 | 495,770,522 | 292,032,037 | 292,032,037 | 292,032,037 | |
| Average length (bases) | 9,031 | 11,605 | 270,806 | 273,490 | 687,624 | 90,745 | 7,043 | 276,545 | 269,900 | 269,402 | |

TruSPAdes v3.6.2 pipeline [17]. Among the three assemblies with PE and MP reads, Assembly 1 was used for subsequent analysis, and gaps were closed with Illumina SLRs by GMcloser (RRID: SCR_000646) [18]. Potentially contaminated sequences were excluded using BLASTN searches against the chloroplast and mitochondrial genome sequences of *Arabidopsis thaliana* (accession numbers NC_000932.1 and NC_001284.2), human genome sequences (hg19 [19]), fungal genome sequences registered with the National Center for Biotechnology Information (NCBI) [20], bacterial genome sequences registered with [21], vector sequences in UniVec [22], and PhiX (NC_001422.1) [23] sequences with *E*-value cutoffs of $1 \times 10^{-10}$ and length coverage >10%. The total length of the resultant assembly (Assembly 5) was 295.7 Mbp.

The results of benchmarking universal single-copy ortholog (BUSCO) analysis (RRID: SCR_015008) [24] identified that 93.1% of BUSCOs were found as complete genes in Assembly 5. We therefore considered that Assembly 5 covered most of the coding regions of the horsegram genome. Sequences shorter than 500 bp were excluded from Assembly 5, and the remaining sequences were designated as MUN_r1.1.

## Linkage map and pseudomolecule construction

To construct chromosome-scale genome sequences, a SNP linkage map was created with the 214 $F_2$ progenies. SNPs segregating in the $F_2$ population were detected by mapping Illumina re-sequence reads of the eight $F_2$ individuals onto the assembled genome using Bowtie2 (RRID: SCR_016368) [25], and by calling variants using SAMtools 0.1.19 (RRID: SCR_002105) [26] and vcftools 0.1.12 (RRID: SCR_001235) [27]. Target amplicon sequencing (TAS) was performed to genotype the identified SNPs according to the methods described in Shirasawa *et al.* [28].

The linkage map was constructed using JoinMap 4 with Kosambi's mapping function (RRID: SCR_009248) [29]. The assembled genome sequence scaffolds were aligned onto the linkage map for pseudomolecule construction. The female parent of the $F_2$ progenies was HPK-4. The male parent was initially considered to be HPKM-193, but this assignment was later found to be wrong when the whole genome sequences of HPK-4, HPKM-193, and the eight $F_2$ progenies were compared. Candidate SNPs segregating in the $F_2$ progenies were

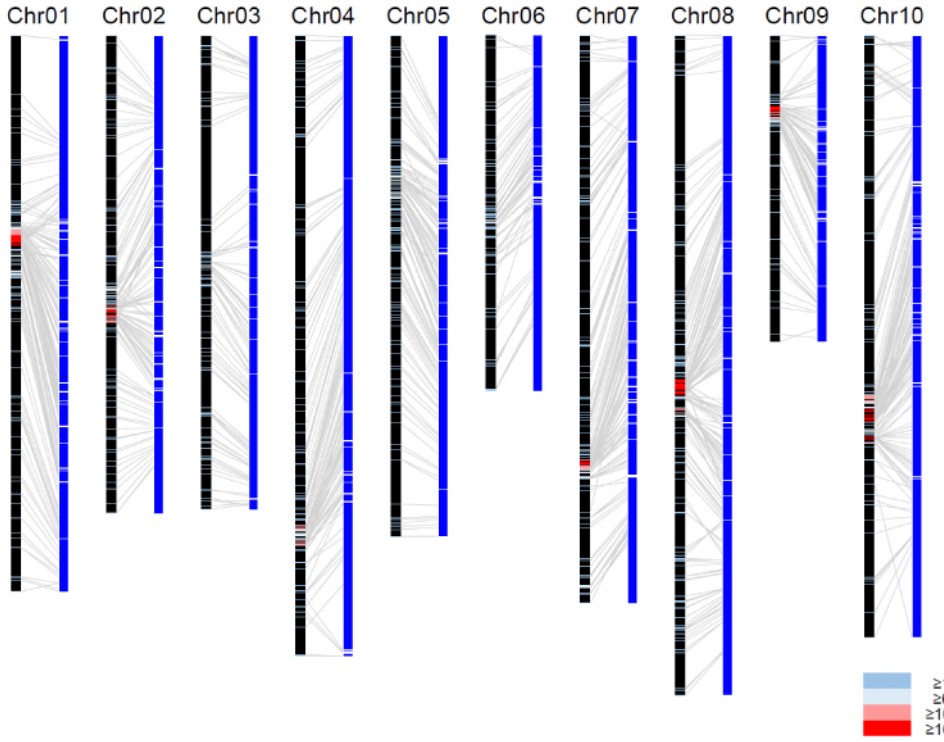

**Figure 2.** Anchoring the horsegram genome assembly to the genetic linkage map. The linkage groups (left black bars) and 219 anchored MUN_r1.1 scaffolds (right blue bars) with 1263 SNPs. The crossbars on the linkage groups show the positions of mapped SNPs. Blue, aqua, pink, and red colors represent the numbers of mapped SNPs per cM of 1–5, 6–10, 10–15, and ≧16, respectively.

identified by mapping the whole genome Illumina sequences of the eight randomly selected $F_2$ progenies onto MUN_r1.1.

A total of 2942 SNPs were identified, and 1378 SNPs were successfully genotyped by TAS analysis in 214 $F_2$ progenies. Of these, 1263 SNPs were mapped onto the ten linkage groups with a total length of 980 cM (Table 3). A total of 219 scaffolds in MUN_r1.1 were then aligned onto the linkage map (Figure 2; Table 3; and in GigaDB [10]). During the process of alignment, two scaffolds were discovered to be misscaffoldings and split. The revised set of scaffolds was designated as MUN_r1.11 (Table 4; Table 2). The number of sequences of MUN_r1.11 was 3,495, with a total length of 294.7 Mbp and an N50 length of 2.8 Mbp. The aligned scaffolds on the linkage map were connected to 10,000 Ns for the construction of chromosome-scale pseudomolecules. The total length of the ten pseudomolecules was 259.2 Mbp, with an N50 length of 28.2 Mbp (Table 4; Table 5). When the total length of the A, G, T, and C bases was compared, the 10 pseudomolecules were found to cover 89% of the scaffolds in MUN_r1.11. The ratios of complete BUSCOs identified in MUN_r1.11 and the 10 scaffolds were 93.1% and 87.4%, respectively. Most of the complete BUSCOs were identified as single copies, suggesting a slow rate of duplication in the coding regions of the assembled genomes.

**Table 3.** Statistics of a SNP linkage map and numbers of anchored scaffolds.

| | Linkage map | | | | Number of anchored scaffolds (MUN_r1.1) |
|---|---|---|---|---|---|
| | Number of mapped SNPs | Length (cM) | Mean distance between SNPs (cM) | Segregation distortion ratio (%) | |
| Chr1 | 148 | 97.2 | 0.66 | 4.05 | 29 |
| Chr2 | 128 | 73.1 | 0.57 | 74.22 | 26 |
| Chr3 | 84 | 120.3 | 1.43 | 3.57 | 14 |
| Chr4 | 131 | 123.1 | 0.94 | 14.50 | 18 |
| Chr5 | 124 | 100.8 | 0.81 | 3.23 | 22 |
| Chr6 | 76 | 88.9 | 1.17 | 14.57 | 16 |
| Chr7 | 148 | 119.1 | 0.80 | 12.84 | 22 |
| Chr8 | 185 | 117.1 | 0.63 | 2.70 | 19 |
| Chr9 | 87 | 51.7 | 0.59 | 81.61 | 25 |
| Chr10 | 152 | 88.8 | 0.58 | 3.95 | 28 |
| Total | 1,263 | 980 | 0.78 | | 219 |

**Table 4.** Statistics on the horsegram genome assembly and CDS.

| | MUN_r1.11 Genome/Scaffolds | MUN_r1.11 Genome/Pseudomolecules | MUN_r1.1_cds CDS |
|---|---|---|---|
| Number of sequences | 3,497 | 10 | 36,105 |
| Total length (bp) | 294,688,765 | 259,245,825 | 38,820,013 |
| Average length (bp) | 84,269 | 25,924,583 | 1,075 |
| Maximum length (bp) | 9,844,273 | 33,386,276 | 15,732 |
| Minimum length (bp) | 500 | 15,505,026 | 150 |
| N50 length (bp) | 2,818,555 | 28,154,654 | 1,488 |
| Total length of AGTC (bp) | 286,841,254 | 255,003,950 | |
| Gaps (bp) | 7,847,511 | 4,241,875 | - |
| GC% | 30.8 | 30.5 | 43.8 |
| Repeat % | 28.99497136 | - | - |
| Number of complete genes | - | - | 35,508 |
| Number of partial genes | - | - | 597 |

**Table 5.** Assembly statistics of MUN_r1.11 pseudomolecules.

| | MUN_chr01 | MUN_chr02 | MUN_chr03 | MUN_chr04 | MUN_chr05 | MUN_chr06 | MUN_chr07 | MUN_chr08 | MUN_chr09 | MUN_chr10 |
|---|---|---|---|---|---|---|---|---|---|---|
| Total length of sequences (bp) | 28,154,654 | 24,194,727 | 23,973,329 | 31,423,847 | 25,354,798 | 18,013,159 | 28,753,260 | 33,386,276 | 15,505,026 | 30,486,749 |
| A | 9,658,539 | 8,292,913 | 8,231,116 | 10,813,335 | 8,677,934 | 6,151,137 | 9,805,582 | 11,440,621 | 5,247,870 | 10,296,716 |
| T | 9,689,700 | 8,302,615 | 8,197,375 | 10,734,237 | 8,667,169 | 6,161,839 | 9,816,935 | 11,432,900 | 5,212,370 | 10,323,063 |
| G | 4,127,487 | 3,524,028 | 3,621,541 | 4,739,623 | 3,794,983 | 2,702,465 | 4,354,257 | 5,051,796 | 2,292,810 | 4,654,132 |
| C | 4,128,504 | 3,558,607 | 3,614,862 | 4,779,430 | 3,830,211 | 2,738,375 | 4,343,032 | 5,030,174 | 2,307,484 | 4,656,183 |
| N | 550,424 | 516,564 | 308,435 | 357,222 | 384,501 | 259,343 | 433,454 | 430,785 | 444,492 | 556,655 |
| Total (ATGC) | 27,604,230 | 23,678,163 | 23,664,894 | 31,066,625 | 24,970,297 | 17,753,816 | 28,319,806 | 32,955,491 | 15,060,534 | 29,930,094 |
| GC% (ATGC) | 29.9 | 29.9 | 30.6 | 30.6 | 30.5 | 30.6 | 30.7 | 30.6 | 30.5 | 31.1 |
| Number of anchored scaffolds | 29 | 26 | 14 | 18 | 22 | 16 | 22 | 19 | 25 | 28 |
| Total length of scaffolds | 27,874,654 | 23,944,727 | 23,843,329 | 31,253,847 | 25,144,798 | 17,863,159 | 28,543,260 | 33,206,276 | 15,265,026 | 30,216,749 |
| Total length of inserted Ns (N10000) | 280,000 | 250,000 | 130,000 | 170,000 | 210,000 | 150,000 | 210,000 | 180,000 | 240,000 | 270,000 |

## Repetitive sequences

Repetitive sequences in the assembled genome were identified by RepeatMasker v3.2.9 (RRID: SCR_012954) [30] for known repetitive sequences registered in Repbase

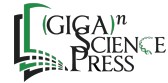

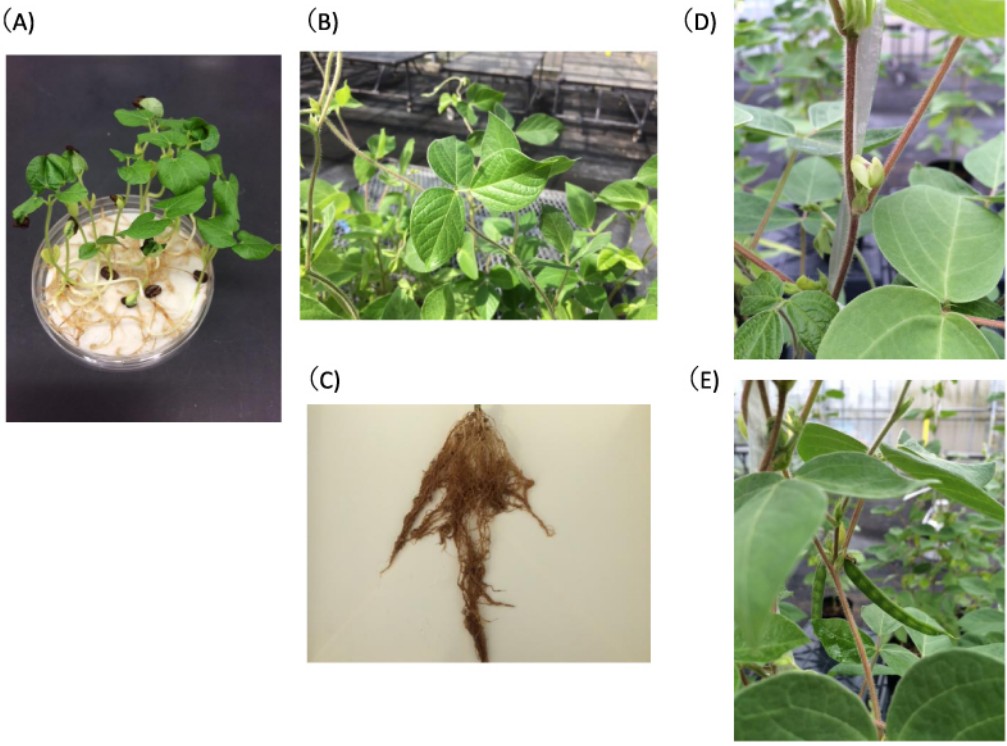

**Figure 3.** Plant materials used for transcript sequences. The seedlings (A), leaves (B), roots (C), flowers (D), and young pods (E) of HPK-4 used for Illumina transcript sequencing.

(RRID: SCR_021169) [31], and *de novo* repetitive sequences were defined by RepeatScout v1.0.5 (RRID: SCR_014653) [32]. A total of 50.2 Mbp of repetitive sequences were identified on the assembled genome, occupying 29% of the total length (Table 6). Of the identified repetitive sequences, the sequences registered in Repbase were found on 12.0% of the assembled genome, whereas unique repetitive sequences, i.e., those not registered in Repbase, were located on 17.0% of the assembled genome. Simple sequence repeat (SSR) motifs were identified by MISA mode in SciRoKo software with the default parameters (RRID: SCR_000941) [33]. A total of 74,362 SSRs were identified in MUN_r1.11 with an average frequency of 0.21 SSR per 100 Kbp [10]. The highest SSR frequency, 0.66 SSR per 100 Kbp, was observed in chr06, and this value was almost three times higher than that in chr03 and chr08 (0.22 SSR per 100 Kbp).

## Transcript sequencing, gene prediction, and annotation

Total RNA of HPK-4 was extracted from seedlings, leaves, roots, flowers, and young pods using the RNeasy Plant Mini Kit (QIAGEN). RNA libraries were constructed by using a TruSeq standard mRNA HT sample prep kit (Illumina). Library sequencing was performed by an Illumina HiSeq system with a read length of 93 nt. Assembly was performed by Trinity [34]. A total of 485 million transcript Illumina reads were obtained from seedlings, leaves, roots, flowers, and young pods of HPK-4 (Figure 3) [10]. *Ab initio* gene prediction was performed by BRAKER1 v1.9 (RRID: SCR_018964) [35] with the obtained transcript sequences. Transposable elements (TEs) were detected by BLASTP searches against the

**Table 6.** Length and ratio of repetitive sequences.

| Category | | MUN_r1.11 bp | MUN_r1.11 %[a] | MUN_chr01 bp | MUN_chr01 %[b] | MUN_chr02 bp | MUN_chr02 %[b] | MUN_chr03 bp | MUN_chr03 %[b] | MUN_chr04 bp | MUN_chr04 %[b] | MUN_chr05 bp | MUN_chr05 %[b] | MUN_chr06 bp | MUN_chr06 %[b] | MUN_chr07 bp | MUN_chr07 %[b] | TSU4_chr08 bp | TSU4_chr08 %[b] | MUN_chr09 bp | MUN_chr09 %[b] | MUN_chr10 bp | MUN_chr10 %[b] |
|---|---|---|---|---|---|---|---|---|---|---|---|---|---|---|---|---|---|---|---|---|---|---|---|
| Known repeats in Repbase — Class I — Interspersed repeats | SINEs | 33,194 | 0.0 | 1,893 | 0.0 | 2,704 | 0.0 | 2,330 | 0.0 | 3,998 | 0.0 | 3,017 | 0.0 | 2,162 | 0.0 | 5,597 | 0.0 | 3,606 | 0.0 | 769 | 0.0 | 3,640 | 0.0 |
| | LINEs | 758,450 | 0.3 | 79,135 | 0.3 | 60,158 | 0.2 | 78,223 | 0.3 | 67,284 | 0.2 | 61,531 | 0.2 | 35,312 | 0.2 | 76,948 | 0.3 | 95,328 | 0.3 | 32,464 | 0.2 | 88,614 | 0.3 |
| | LTR elements — Total | 18,946,454 | 6.4 | 2,502,160 | 8.9 | 1,905,720 | 7.9 | 1,430,860 | 6.0 | 1,626,130 | 5.2 | 1,474,150 | 5.8 | 938,839 | 5.2 | 1,864,330 | 6.5 | 2,080,920 | 6.2 | 1,268,235 | 8.2 | 1,796,168 | 5.9 |
| | LTR elements — Copia | 12,243,368 | 4.2 | 1,611,130 | 5.7 | 1,257,510 | 5.2 | 902,900 | 3.8 | 1,167,090 | 3.7 | 949,455 | 3.7 | 658,107 | 3.6 | 1,176,390 | 4.1 | 1,417,100 | 4.2 | 740,132 | 4.8 | 1,173,567 | 3.8 |
| | LTR elements — Gypsy | 6,273,336 | 2.1 | 820,553 | 2.9 | 601,000 | 2.5 | 500,201 | 2.1 | 417,277 | 1.3 | 496,833 | 2.0 | 254,838 | 1.4 | 654,972 | 2.3 | 613,122 | 1.8 | 491,823 | 3.2 | 594,170 | 1.9 |
| Class II | DNA elements | 3,100,519 | 1.1 | 380,164 | 1.4 | 280,403 | 1.2 | 227,524 | 0.9 | 294,636 | 0.9 | 206,575 | 0.8 | 121,303 | 0.7 | 294,742 | 1.0 | 307,336 | 0.9 | 198,994 | 1.3 | 289,814 | 1.0 |
| | Unclassified | 420 | 0.0 | 0 | 0.0 | 77 | 0.0 | 0 | 0.0 | 0 | 0.0 | 0 | 0.0 | 0 | 0.0 | 0 | 0.0 | 243 | 0.0 | 0 | 0.0 | 63 | 0.0 |
| | Helitrons | 182,492 | 0.1 | 24,503 | 0.1 | | | 15,763 | 0.1 | 8,828 | 0.0 | 12,915 | 0.1 | 16,031 | 0.1 | 8,133 | 0.0 | 17,183 | 0.1 | 8,945 | 0.1 | 13,311 | 0.0 |
| | Low complexity | 3,615,499 | 1.2 | 222,186 | 0.8 | | | 203,104 | 0.8 | 238,959 | 0.8 | 194,268 | 0.8 | 134,128 | 0.7 | 269,659 | 0.9 | 257,639 | 0.8 | 119,470 | 0.8 | 256,405 | 0.8 |
| | Simple repeat | 8,038,181 | 2.7 | 744,057 | 2.6 | | | 581,825 | 2.4 | 793,431 | 2.5 | 634,665 | 2.5 | 493,067 | 2.7 | 687,667 | 2.4 | 783,434 | 2.4 | 367,990 | 2.4 | 753,700 | 2.5 |
| | Unknown | 13,270 | 0.0 | 366 | 0.0 | | | 642 | 0.0 | 1,053 | 0.0 | 520 | 0.0 | 1,094 | 0.0 | 1,491 | 0.0 | 884 | 0.0 | 859 | 0.0 | 2,820 | 0.0 |
| | Subtotal | 35,247,906 | 12.0 | 4,044,950 | 14.4 | | 13.3 | 2,581,448 | 10.8 | 3,088,969 | 9.8 | 2,627,752 | 10.4 | 1,775,449 | 9.8 | 3,250,692 | 11.3 | 3,614,610 | 10.8 | 2,038,420 | 13.1 | 3,241,129 | 10.6 |
| Unique repeats | Unknown | 49,554,792 | 16.8 | 6,234,995 | 22.1 | | 19.3 | 3,435,357 | 14.3 | 4,094,114 | 13.0 | 3,347,931 | 13.2 | 2,243,739 | 12.4 | 4,226,300 | 14.7 | 5,021,333 | 15.1 | 3,433,883 | 22.1 | 5,105,098 | 16.7 |
| | Simple repeat | 642,225 | 0.2 | 67,981 | 0.2 | | 0.2 | 47,815 | 0.2 | 66,279 | 0.2 | 51,673 | 0.2 | 40,270 | 0.2 | 61,328 | 0.2 | 65,386 | 0.2 | 31,004 | 0.2 | 61,449 | 0.2 |
| | Subtotal | 50,197,017 | 17.0 | 6,302,976 | 22.4 | | 19.6 | 3,483,172 | 14.5 | 4,160,393 | 13.3 | 3,399,604 | 13.4 | 2,284,009 | 12.7 | 4,287,628 | 14.9 | 5,086,719 | 15.3 | 3,464,887 | 22.3 | 5,166,547 | 16.9 |
| Total | | 85,444,923 | 29.0 | 10,347,926 | 36.8 | 7,960,760 | 32.9 | 6,064,620 | 25.3 | 7,249,362 | 23.1 | 6,027,356 | 23.8 | 4,059,458 | 22.5 | 7,538,320 | 26.2 | 8,701,329 | 26.1 | 5,503,307 | 35.5 | 8,407,676 | 27.6 |

[a] Primarily poly-purine/poly-pyrimidine stretches, or regions of extremely high AT or GC content. Stretches of DNA (100 bp) were masked when they were >87% AT or >89% GC, and 30 bp stretches were masked when they contained 29 A/T (or GC) nucleotides.

[b] N bases were excluded from the calculation.

**Table 7.** Statistics of candidate genes predicted by BRAKER1 v1.9.

|  | All predicted genes | MUN_r1.1_cds |
|---|---|---|
|  |  | Exclude TE, pseudo and short genes |
| Number of sequences | 46,095 | 36,105 |
| Total length (bp) | 48,277,179 | 38,820,013 |
| Average length (bp) | 1,047 | 1,075 |
| Max length (bp) | 15,732 | 15,732 |
| Min length (bp) | 60 | 150 |
| N50 length (bp) | 1,443 | 1,488 |
| GC% | 43.3 | 43.8 |

**Table 8.** Number of CDSs showing significant similarity by BLASTP and domain searches against NCBI NR and InterPro.

| Number of CDSs | % to Total | | Classification | Tag | | All predicted genes | MUN_r1.1_cds |
|---|---|---|---|---|---|---|---|
|  | All predicted genes | MIN_r1.1_cds |  | Similarity against NR | Domain |  |  |
| 19,874 | 43 | 55 | Complete | f | d | Included | Included |
| 1554 | 3 | 4 | Complete | f | - | Included | Included |
| 3574 | 8 | 10 | Complete | p | d | Included | Included |
| 2996 | 6 | 8 | Complete | p | - | Included | Included |
| 1052 | 2 | 3 | Complete | - | d | Included | Included |
| 6458 | 14 | 18 | Complete | - | - | Included | Included |
| 26 | 0 | 0 | Partial | f | d | Included | Included |
| 17 | 0 | 0 | Partial | f | - | Included | Included |
| 49 | 0 | 0 | Partial | p | d | Included | Included |
| 73 | 0 | 0 | Partial | p | - | Included | Included |
| 124 | 0 | 0 | Partial | - | d | Included | Included |
| 308 | 1 | 1 | Partial | - | - | Included | Included |
| 126 | 0.3 | - | Pseudo |  |  | Included | Not included |
| 107 | 0.2 | - | Short |  |  | Included | Not included |
| 9757 | 21.2 | - | TE |  |  | Included | Not included |

BLASTP Search against NCBI NR database
f: $E$-value $\leq 1 \times 10^{-20}$ and similarity $\geq 70\%$
p: $E$-value $\leq 1 \times 10^{-20}$ and similarity $< 70\%$
InterProScan against the InterPro database
d: $E$-value $\leq 1.0$

NCBI NR protein database [36] with an $E$-value cutoff of $1 \times 10^{-10}$. Domain search was performed by InterProScan against the InterPro database with an $E$-value cutoff of 1.0 (RRID: SCR_005829) [37].

A total of 46,095 gene sequences were predicted on the assembled genome with a total length of 48.3 Mbp (Table 7). After removal of TEs and both pseudo and short gene sequences, 36,105 gene sequences remained, and this set of sequences was designated as MUN_r1.1_cds (Table 4). The ratio of complete BUSCOs identified on MUN_r1.1_cds was 91.2%. Of the 36,105 sequences, 35,508 were classified as complete genes and 597 as partial. The coding sequences (CDSs) were further tagged with "f" (full similarity), "p" (partial similarity), and "d" (domain) according to the similarity level against the non-redundant database (f: $E$-values $\leq 1 \times 10^{-20}$ and identity $\geq 70\%$; p: $E$-values $\leq 1 \times 10^{-20}$ and identity $< 70\%$) and the InterPro database (d: $E$-values $\leq 1.0$; Table 8). Of the 36,105 sequences, 21,471 (59.4%) were tagged with "f" and 6,692 (18.5%) with "p". The number of gene sequences tagged with "d" was 24,575 (68.1%).

**Table 9.** Numbers of putative tRNA and rRNA encoding genes identified in MUN_r1.1 and other legume species.

| tRNA | | | | | |
|---|---|---|---|---|---|
| **Encode** | ***M. uniflorum* (MUN_r1.11)** | ***P. vulgaris* (Pvulgaris_218_v1.0)** | ***V. angularis* (Vangularis_v1.genome)** | ***L. japonicus* (Lj3.0_pseudomol)** | ***A. thaliana* (TAIR10_genome)** |
| Ala | 40 | 41 | 44 | 40 | 33 |
| Arg | 34 | 43 | 52 | 54 | 39 |
| Asn | 17 | 22 | 30 | 28 | 19 |
| Asp | 27 | 30 | 32 | 32 | 28 |
| Cys | 12 | 16 | 20 | 76 | 17 |
| Gln | 21 | 21 | 24 | 23 | 19 |
| Glu | 31 | 28 | 32 | 40 | 27 |
| Gly | 46 | 51 | 44 | 48 | 43 |
| His | 13 | 15 | 14 | 19 | 12 |
| Ile | 101 | 30 | 32 | 29 | 25 |
| Leu | 49 | 56 | 53 | 57 | 45 |
| Lys | 34 | 43 | 35 | 41 | 33 |
| Met | 38 | 39 | 43 | 48 | 31 |
| Phe | 21 | 28 | 20 | 22 | 17 |
| Pro | 43 | 48 | 36 | 46 | 68 |
| Ser | 50 | 50 | 51 | 44 | 72 |
| Thr | 690 | 28 | 24 | 36 | 26 |
| Trp | 15 | 16 | 17 | 21 | 16 |
| Tyr | 15 | 17 | 18 | 22 | 83 |
| Val | 36 | 41 | 37 | 34 | 32 |
| Subtotal | 669 | 663 | 658 | 760 | 685 |
| Subtotal (%) | 97.0 | 97.4 | 98.7 | 88.6 | 98.0 |
| Pseudo | 19 | 12 | 7 | 80 | 13 |
| SeC | 0 | 1 | 0 | 12 | 0 |
| Sup | 0 | 1 | 0 | 0 | 0 |
| Undet | 2 | 4 | 2 | 6 | 1 |
| Total | 690 | 681 | 667 | 858 | 699 |
| **rRNA** | | | | | |
| **Encode gene** | ***M. uniflorum* (MUN_r1.1)** | ***P. vulgaris* (Pvulgaris_218_v1.0)** | ***V. angularis* (Vangularis_v1.genome)** | ***L. japonicus* (Lj3.0_pseudomol)** | ***A. thaliana* (TAIR10_genome)** |
| 18S | 40 | 48 | 224 | 27 | 4 |
| 25S | 87 | 83 | 421 | 64 | 5 |
| 5.8S | 12 | 8 | 139 | 6 | 2 |
| Total | 139 | 139 | 784 | 97 | 11 |

[a]Pseudo, SeC, Sup, and Undet represent pseudogenes, selenocysteine tRNAs, possible suppressor tRNAs, and tRNAs with undetermined/unknown isotypes, respectively.
[b]Accession numbers used for identification or 5.8S, 18S, and 25S rRNAs were X52320.1, X16077.1, and X52320.1, respectively.

Transfer RNA genes were predicted using tRNAscan-SE ver. 1.23 with the default parameters [38], and compared with the numbers on the genomes of *Phaseolus vulgaris* (Pvulgaris_218_v1.0, 681) [39], *Vigna angularis* (Vangularis_v1.a1) [40], *Lotus japonicus* (Lj3.0) [41], and *A. thaliana* (Araport11 [42]). Total number of putative tRNA genes in the assembled genomes (MUN_r1.11) was 690, almost the same as the numbers for the genomes of *P. vulgaris* (681), *V. angularis* (667), and *A. thaliana* (699, Table 9). rRNA genes were predicted by BLAST searches (*E*-value cutoff of $1 \times 10^{-10}$) with query sequences of *A. thaliana* 5.8S and 25S rRNAs (X52320.1) and 18S rRNA (X16077.1). The total number of putative rRNA genes identified in the genome was 139, which was again the same as the number in the *P. vulgaris* genome.

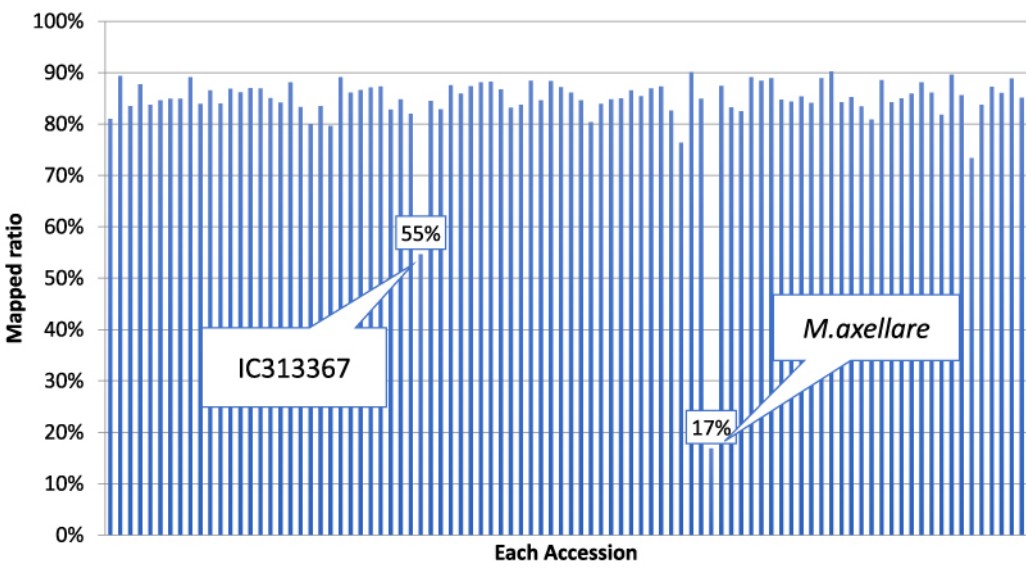

**Figure 4.** Mapped ratios of the dd-RAD-Seq reads of 92 accessions.

## Diversity analysis in genetic resources

Only two species in the genus *Macrotyloma*, i.e., horsegram and *M. geocarpum*, are used as crops. It was speculated that horsegram domestication occurred in India twice: once in northwestern India at 4000 years before present, and once on the Indian Peninsula at 3500 years before present [43]. In addition, horsegram has narrow genetic diversity, as revealed by molecular analysis [44]. The genetic diversity of 91 cultivated horsegram accessions and one *M. axellare* accession, a wild relative of horsegram that is maintained at CSK-HPAU, were investigated based on dd-RAD-Seq analysis [10]. Library construction and variant calling were performed according to Shirasawa *et al.* [45]. The ddRAD-Seq reads were generated by an Illumina HiSeq 2000 system with a read length of 93 nt and mapped onto the assembled genome sequences. The two accessions, IC139449 and IC547543, were excluded from further analysis because of the small number of obtained reads. The mapped ratio of the reads onto the genome (MUN_r1.11) ranged from 80–90% in most of the accessions (Figure 4). However, *M. axellare* and one horsegram accession (IC313367) showed low mapping ratios of 17% and 55%, respectively. *M. axellare* was excluded from further analysis because of its low mapping ratio.

A total of 277 SNPs were identified in the remaining 89 accessions across the genome [10]. The Jaccard similarity coefficients of the 277 SNPs were calculated using GGT 2.0 [46], and a neighbor-joining (NJ) phylogenetic tree was constructed using MEGA ver 10.1.8 (RRID: SCR_000667) [47]. The NJ tree classified the 89 accessions into two clusters (Figure 5). Cluster 1 included varieties bred in the CSK-HPAU, which are prefixed with "HPK". Most of the HPK varieties showed very close genetic relations and formed a subcluster (HPK cluster); the single exception was HPK-4.

## Whole genome structure in horsegram

Figure 6 shows a graphical view of the horsegram genome structure with a graph drawn by Circos (Figure 6; RRID: SCR_011798) [48]. Repetitive sequences were frequently observed in

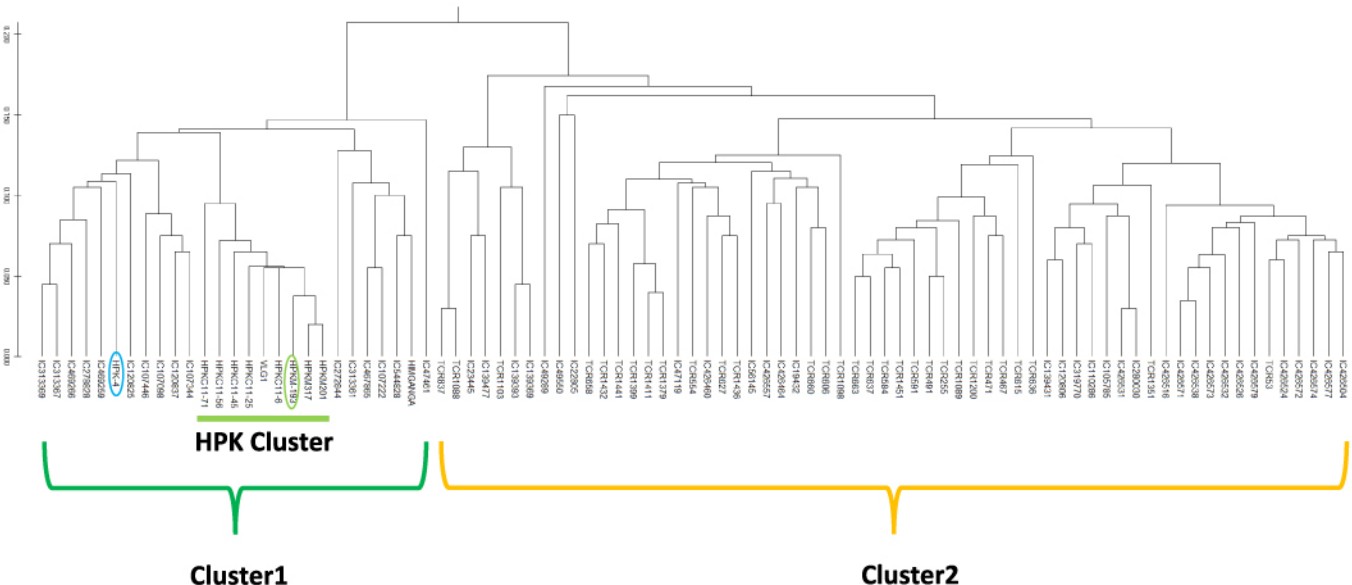

**Figure 5.** A phylogenetic tree of the 89 horsegram accessions based on 277 SNPs. HPK-4, used in the reference genome construction; HPK-4, used in the reference genome construction, and HPKM-193; the obtained whole genome sequences of the accessions are circled with blue and green lines, respectively.

the midsection of each chromosome, and the tendency was more pronounced in horsegram-specific sequences (Figure 6A). The ratio of repetitive sequences commonly observed in all five species was quite low, suggesting the uniqueness of repetitive sequences compared to the gene sequences. The gene sequences commonly observed between horsegram and the other compared species tended to be distributed to the two end regions of the chromosomes (Figure 6B). On the other hand, horsegram-specific gene sequences were distributed more uniformly across the genome, suggesting the unique structure of the horsegram genome.

Copy number variations (CNVs) of one horsegram accession, HPKM-193, were detected against the HPK-4 genome (Figure 6C) based on the whole genome sequence reads of HPKM-193 using CNV-Seq (RRID: SCR_013357) [49] with a 1-Mbp window. CNVs with a minus log2 ratio were particularly observed on chr09 and chr02.

Of the 277 SNPs identified among the 89 horsegram accessions, 255 were located across the genome sequences of 10 chromosomes (Figure 6D). In each chromosome, SNPs were mostly identified in the regions where common putative genes of horsegram and the other compared species were located, particularly for chr04, chr07, chr08, and chr10. The differing trends in variable distribution is thought to reflect the presence of varying degrees of selection pressure in the horsegram germplasm resources in Himachal Pradesh.

SNP density mapped on the linkage map is illustrated in Figure 6E. As in the case of the CNVs, distribution bias was observed in the SNPs of HPKM-193; however, this bias was not like that in CNVs. A higher SNP density was observed in the midsection in most of the chromosomes. Chr06 showed less variation than the other chromosomes.



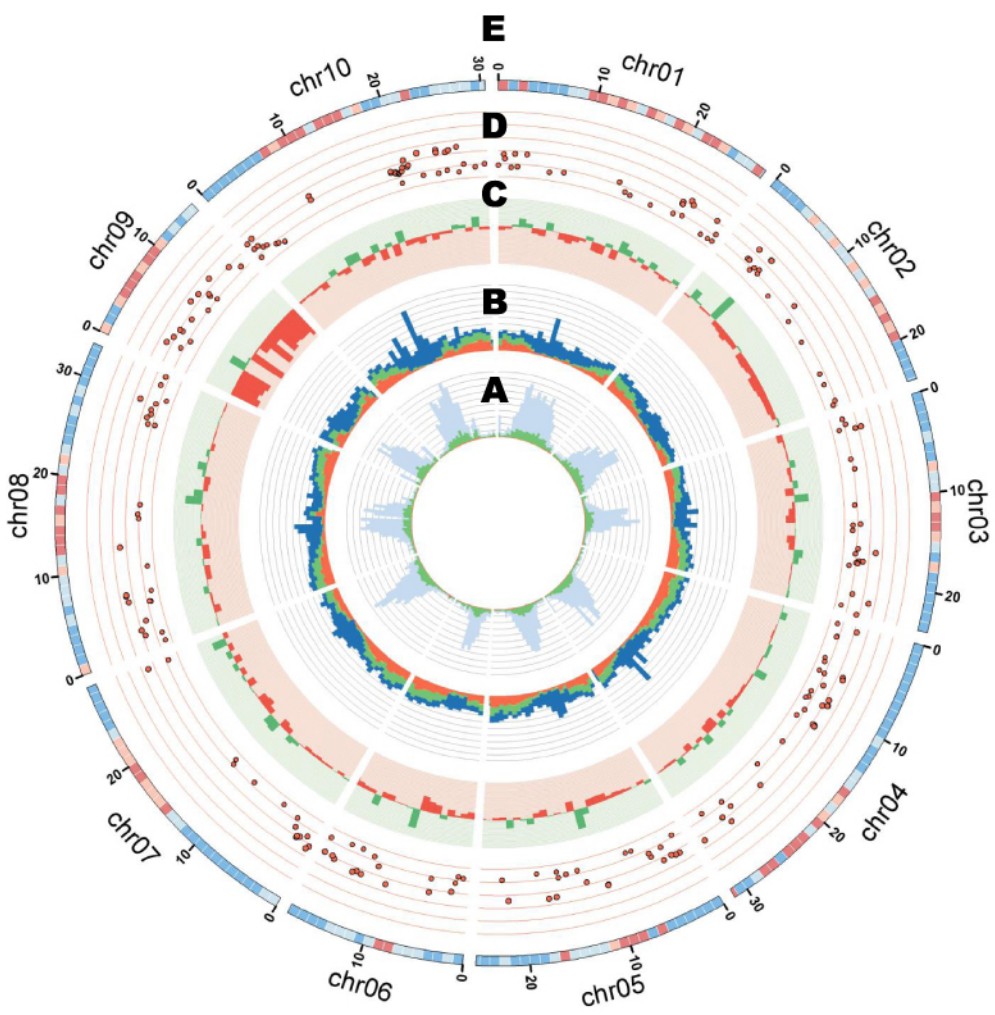

**Figure 6.** Graphical view of the horsegram genome structure. **(A)** Ratios of repetitive sequences in 1-Mbp windows. Blue bars represent horsegram-specific sequences. Green bars show sequences commonly observed in horsegram and three other legume species: *P. vulgaris*, *V angularis*, and *L. japonicus*. Red bars show sequences commonly observed in the four legume species and *A. thaliana*. **(B)** Numbers of the predicted horsegram genes (MUN_r1.1_cds) in 1-Mbp windows. The bar colors are the same as in (A). **(C)** CNV distribution in a 1-Mbp window. Green and red dots show log2 ratio plus and minus values, respectively. **(D)** Pi values and positions of the 255 SNPs identified in the 89 horsegram accessions by using dd-RAD-Seq. The distance between horizontal lines represents a Pi value of 0.1. **(E)** SNP density identified among the $F_2$ mapping population based on whole genome re-sequencing of the eight $F_2$ progenies. Blue, pale blue, pale pink, and pink indicate numbers of SNPs $\leqq 50$, $\leqq 100$, $\leqq 150$, and $150 <$ in 1-Mbp windows, respectively.

## Genes related to drought tolerance

Horsegram is considered one of the most drought-tolerant legume crop species. Personal investigation showed that plants can survive for more than 20 days without water under controlled conditions. A study by Bhardwaj *et al.* [50] described a transcriptome analysis of eight shoot and root tissues of a drought-sensitive (M-191) genotype and a drought-tolerant (M-249) genotype of horsegram under controlled and drought stress conditions. This study identified some important genic regions responsible for drought tolerance.

To estimate genes related to drought tolerance in the horsegram genome, a BLASTP search of the 36,105 putative genes was performed against amino acid sequences of

*A. thaliana* (Araport11), and hit genes were further used in BLAST searches against DroughtDB [51], the NCBI NR protein database, and Plant Stress Gene Database [52]. A total of 158 horsegram genes showed significant similarity to the 78 genes in DroughtDB [10]. The most frequently hit gene was ABCG40, which encodes a protein that functions as an ABC transporter, and showed significant similarity to 14 horsegram genes. OST1/SRK2E and AtrbohF were also frequently identified, with hits to seven and six horsegram genes, respectively. Of the 158 genes, 93 showed the same domain sequences as the *A. thaliana* gene, and 52 were like the genes registered in the PSGD. These genes were indicated to have a greater likelihood of being candidate genes related to drought tolerance.

## Comparative and phylogenetic analyses with other legume species

Horsegram belongs to the subtribe *Phaseolinae* in the millettioid clade, along with *P. vulgaris* and *V. angularis*. The genome structure of horsegram was compared with those of *P. vulgaris* (Pvulgaris_218_v1.0), *V. angularis* (Vangularis_v1.a1), and *L. japonicus* (Lr_3.0).

The predicted gene sequences in MUN_r1.1_cds were clustered with other plant species (*P. vulgaris*, *V. angularis*, *L. japonicus*, and *A. thaliana*) for comparison at the protein sequence level. A total of 73,457 clusters were generated using the program CD-HIT (RRID: SCR_007105) [53] (Table 10). Of the 36,105 putative gene sequences, 21,369 (59.2%) genes were clustered with other plant species and 14,736 (40.8%) were considered horsegram-specific genes (Figure 7). A total of 3738 (10.4%) horsegram gene sequences were clustered with 3,864 *P. vulgaris* and 3,713 *V. angularis* genes, which were considered millettioid-specific genes. Common genes in legumes were identified for 6550 (18.1%) horsegram gene sequences, based on clusters with *P. vulgaris*, *V. angularis*, and *L. japonicus*.

Functional analysis was performed for MUN_r1.1_cds by classifying 36,105 putative genes into the Gene Ontology (GO) and euKaryotic clusters of Orthologous Groups (KOG) databases [54]. A total of 24,699 (68.4%) putative genes were annotated with GO categories including 9086 (25.2%) genes involved in biological processes, 4127 (11.4%) genes coding for cellular components, and 1377 (38.7%) genes associated with molecular functions (Figure 8). The ratio of annotated horsegram genes was smaller than those of the other species. The species with a ratio of classified GO categories most like that of horsegram was *L. japonicus*. A total of 18,630 (51.6%) putative genes showed significant similarity to genes in the KOG database (Figure 9). As in the results for GO, the ratio of hit genes was lower than for the other four species.

Clear relationships were observed with a warm-season legume, *V. angularis*, and one-on-one relationships were observed between horsegram chr02 (Mun_chr02) and *V. angularis* chr09 (Va_chr09), Mun_chr04 and Va_chr02, Mun_chr06 and Va_chr10, Mun_chr07 and Va_chr08, Mun_chr08 and Va_chr04, and Mun_chr09 and Va_chr05 (Figure 10A). The syntenic relations with *P. vulgaris* were slightly more complex than those with *V. angularis*, and those with the cool-season legume *L. japonicus* were more fragmented.

Synonymous substitutions per site (Ks) were estimated by comparing gene pairs in each combination of horsegram, *P. vulgaris*, *V. angularis*, *L. japonicus*, and *A. thaliana* (Araport11) by KaKs Calculator [55] based on the clustered genes using the CD-HIT program (Figure 10B). The similar distributions of horsegram, *P. vulgaris*, and *V. angularis* indicated the close relations among the three species. The ratios of gene pairs showing Ks values less than 0.1% were 21.1% between horsegram and *P. vulgaris* and 8.4% between horsegram



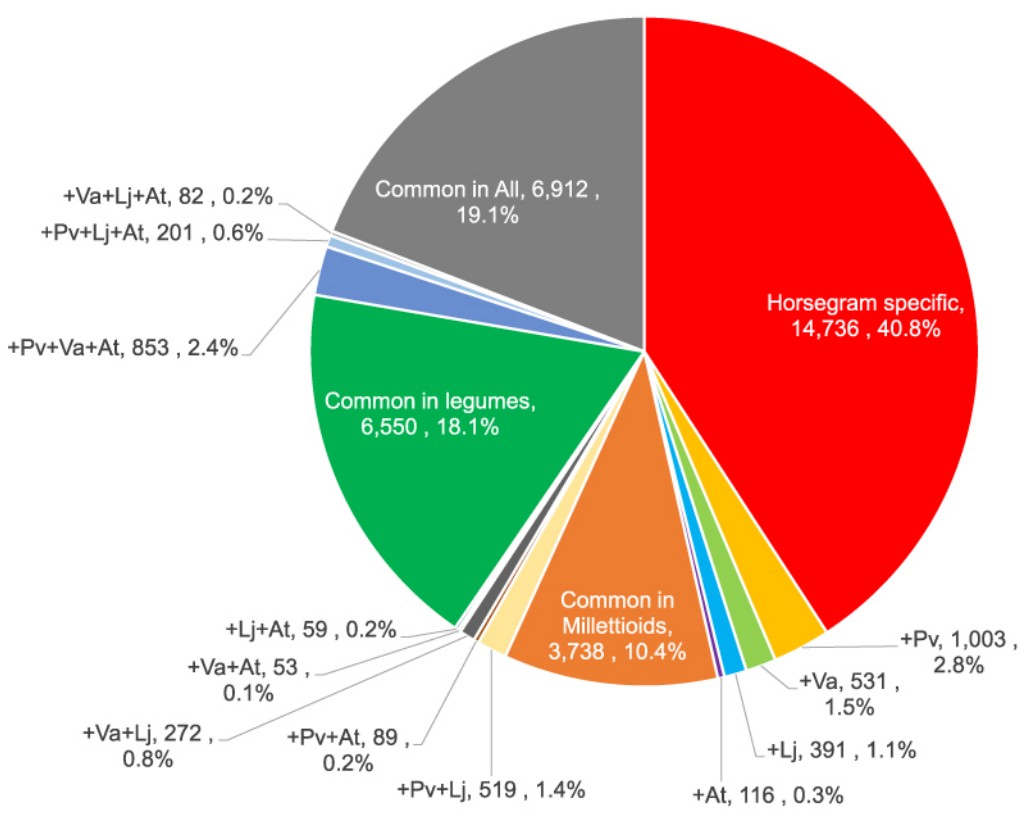

**Figure 7.** Ratios of genes of horsegram (MUN_r1.1_cds) clustered with those of four other plant species. Pv, Va, Li, and At represent genes of *P. vulgaris* (Pvulgaris_218_v1.0), *V. angularis* (Vangularis_v1.a1), *L. japonicus* (Lj_r3.0), and *A. thaliana* (Araport11), respectively.

and *V. angularis*, suggesting that there was a closer relationship between horsegram and *P. vulgaris* at the gene level.

The phylogenetic analysis was performed with *Medica truncatula* (r5.0) [56] and *Glycine max* (Glma4 [57]) in addition to *P. vulgaris, V. angularis, L. japonicus,* and *A. thaliana*. A total of 978 common single-copy genes were identified for horsegram and the six species by clustering the genes using OrthoFinder (RRID: SCR_017118) [58]. Multiple alignment was performed for the 978 single-copy genes using Muscle (RRID: SCR_011812) [59], and gaps were excluded by Gblock [60]. An NJ tree was created with the 4154 single-copy orthologous genes identified in the four legume species by MEGA 7.0.9 beta (RRID: SCR_000667) [61] and TIMETREE (RRID: SCR_021162) [62]. *A. thaliana* was used for the outgroup (Figure 10C). When the divergence time between *M. truncatula* and *G. max* was considered to be 53 million years ago, it was estimated that horsegram diverged from *P. vulgaris* and *V. angularis* 20.75 million years ago (Figure 10C). Among the four legume species in millettioids, *P. vulgaris* and *V. angularis* shared closer relations with each other than with horsegram, and horsegram was closer to *P. vulgaris* and *V. angularis* than to *G. max*. The results are in consonance with a previous study based on a comparison of eight chloroplast regions [63].

**Table 10.** Number of gene clusters in horsegram and the four plant species, *P. vulgaris* (Pvulgaris_218_v1.0), *V. angularis* (Vangularis_v1.a1), *L. japonicus* (Lj_r3.0), and *A. thaliana* (Araport11).

| Clustered species | Number of clustered species | Number of clusters | Number of clustered genes | | | | |
|---|---|---|---|---|---|---|---|
| | | | Horsegram | *P. vulgaris* (Pv) | *V. angularis* (Va) | *L. japonicus* (Lj) | *A. thaliana* (At) |
| Horsegram | 1 | 9,578 | 14,736 | 0 | 0 | 0 | 0 |
| *P. vulgaris* (Pv) | 1 | 3,449 | 0 | 4,114 | 0 | 0 | 0 |
| V. angularis (Va) | 1 | 8,306 | 0 | 0 | 9,545 | 0 | 0 |
| *L. japonicus* (Lj) | 1 | 17,544 | 0 | 0 | 0 | 21,677 | 0 |
| *A. thaliana* (At) | 1 | 15,177 | 0 | 0 | 0 | 0 | 18,129 |
| Horsegram + Pv | 2 | 897 | 1,003 | 1,028 | 0 | 0 | 0 |
| Horsegram + Va | 2 | 471 | 531 | 0 | 530 | 0 | 0 |
| Horsegram + Lj | 2 | 362 | 391 | 0 | 0 | 484 | 0 |
| Horsegram + At | 2 | 111 | 116 | 0 | 0 | 0 | 147 |
| Pv + Va | 2 | 1,031 | 0 | 1,248 | 1,159 | 0 | 0 |
| Pv + Lj | 2 | 262 | 0 | 293 | 0 | 333 | 0 |
| Pv + At | 2 | 88 | 0 | 97 | 0 | 0 | 130 |
| Va + Lj | 2 | 290 | 0 | 0 | 317 | 382 | 0 |
| Va + At | 2 | 92 | 0 | 0 | 95 | 0 | 113 |
| Lj + At | 2 | 326 | 0 | 0 | 0 | 398 | 415 |
| Horsegram + Pv + Va (Common in Millettioids) | 3 | 3,189 | 3,738 | 3,864 | 3,713 | 0 | 0 |
| Horsegram + Pv + Lj | 3 | 458 | 519 | 521 | 0 | 594 | 0 |
| Horsegram + Pv + At | 3 | 87 | 89 | 92 | 0 | 0 | 110 |
| Horsegram + Va + Lj | 3 | 230 | 272 | 0 | 255 | 292 | 0 |
| Horsegram + Va + At | 3 | 49 | 53 | 0 | 54 | 0 | 58 |
| Horsegram + Lj + At | 3 | 56 | 59 | 0 | 0 | 72 | 74 |
| Pv + Va + Lj | 3 | 581 | 0 | 649 | 664 | 725 | 0 |
| Pv + Va + At | 3 | 94 | 0 | 101 | 99 | 0 | 118 |
| Pv + Lj + At | 3 | 66 | 0 | 79 | 0 | 79 | 89 |
| Va + Lj + At | 3 | 41 | 0 | 0 | 45 | 51 | 53 |
| Horsegram + Pv + Va + Lj (common in legumes) | 4 | 5,011 | 6,550 | 6,591 | 6,472 | 6,847 | 0 |
| Horsegram + Pv + Va + At | 4 | 670 | 853 | 857 | 845 | 0 | 873 |
| Horsegram + Pv + Lj + At | 4 | 173 | 201 | 196 | 0 | 222 | 208 |
| Horsegram + Va + Lj + At | 4 | 76 | 82 | 0 | 83 | 102 | 98 |
| Pv + Va + Lj + At | 4 | 302 | 0 | 370 | 365 | 420 | 385 |
| Common in all | 5 | 4,390 | 6,912 | 7,097 | 7,000 | 7,056 | 6,638 |
| Number of genes | | | 36,105 | 27,197 | 31,241 | 39,734 | 27,621 |
| Number of clustered genes | | | 36,105 | 27,197 | 31,241 | 39,734 | 27,638 |
| Number of non-clustered genes | | | 0 | 0 | 0 | 0 | 17 |

## Data validation and quality control

The quality of assembled genome and gene sequences was investigated by using 1375 Embryophyta BUSCOs (v 3.0, obd10; RRID: SCR_015008) [24]. A total of 1340 (93.1%) BUSCOs were identified on the assembled scaffolds (MUN_r1.11), while 1259 (87.4%) and 1313 (91.2%) were converted by the pseudomolecules and CDS sequences (Table 11).

## Reuse potential

In this study, we have provided a first-draft genome assembly of horsegram cultivar (HPK-4) and investigated features of the horsegram genome and gene sequences as well as the genetic diversity of the accessions. This information will help to establish an efficient breeding program for horsegram by integrating conventional breeding with marker-based biotechnological tools. Finally, the genomic information revealed in this study can be applied to the improvement of other disadvantageous food legumes.



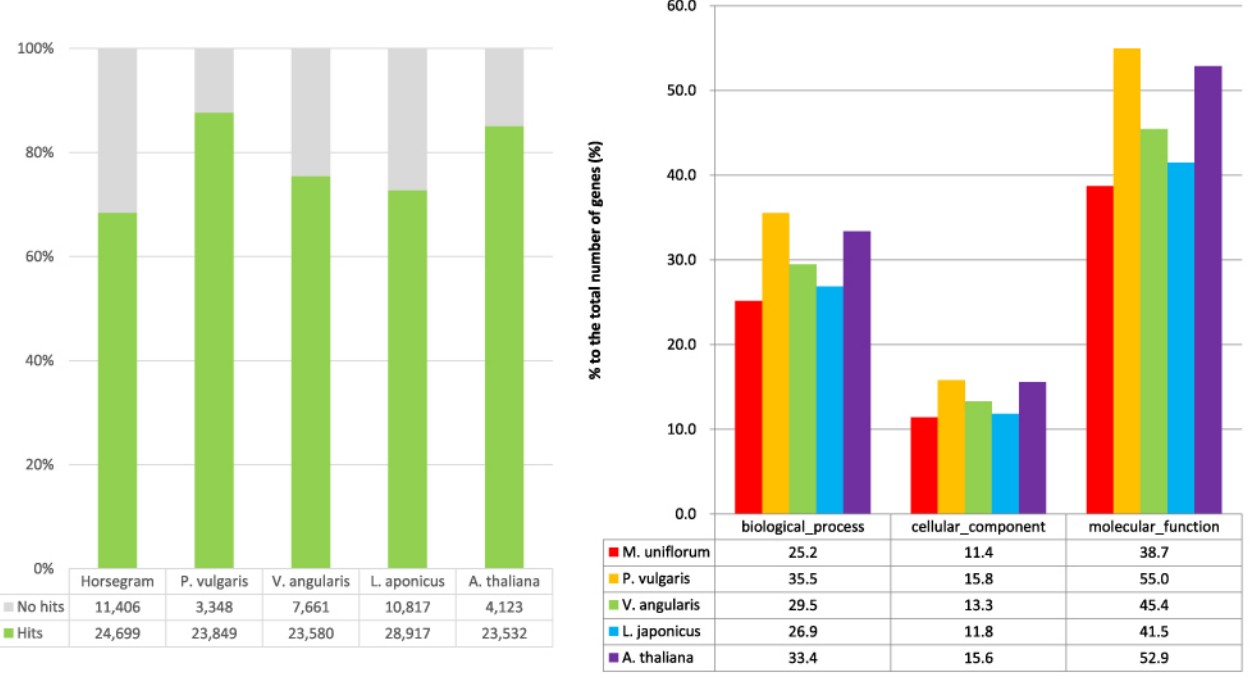

**Figure 8.** Comparison of genes annotated by the GO database in horsegram (MUN_r1.1_cds), *P. vulgaris* (Pvulgaris_218_v1.0), *V. angularis* (Vangularis_v1.a1), *L. japonicus* (Lj_r3.0), and *A. thaliana* (Araport11). (**A**) Numbers and ratios of genes annotated by GO database. (**B**) Ratios of the classified GO categories in the predicted genes.

**Table 11.** Statistics of the horsegram genome assembly and CDS.

|  | MUN_r1.11 Genome/Scaffolds | MUN_r1.11 Genome/Pseudomolecules | MUN_r1.1_cds CDS |
|---|---|---|---|
| BUSCOs |  |  |  |
| Complete | 1340 (93.1%) | 1259 (87.4%) | 1313 (91.2%) |
| Complete single-copy | 1252 (86.9%) | 1181 (82%) | 1208 (83.9%) |
| Complete duplicated | 88 (6.1%) | 78 (5.4%) | 105 (7.3%) |
| Fragmented | 26 (1.8%) | 31 (2.2%) | 23 (1.6%) |
| Missing | 74 (5.1%) | 150 (10.4%) | 104 (7.2%) |

## DATA AVAILABILITY

The genome assembly data, annotations, and gene models are available at the Horsegram Database [64]. The obtained genome sequence reads are available from the DNA Databank of Japan (DDBJ) Sequence Read Archive (DRA) under the BioProject accession number PRJDB5374.

Data sets supporting the results of this article are available in *GigaScience* Database [10].

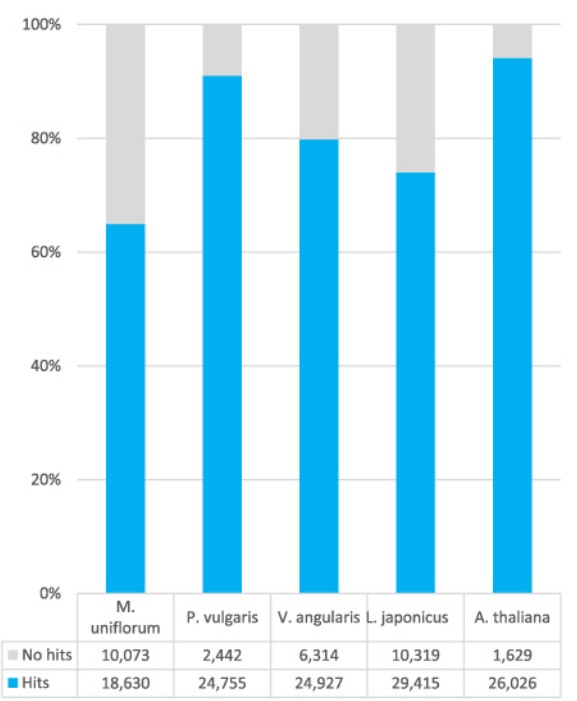

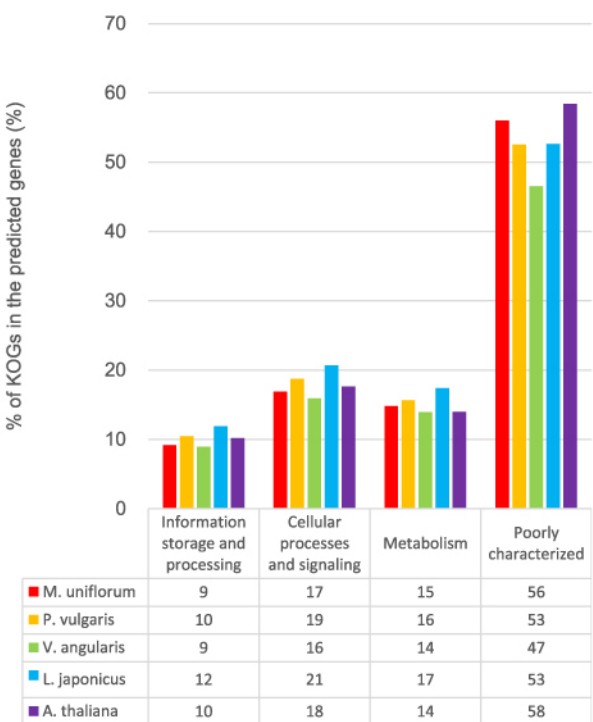

**Figure 9.** Comparison of genes annotated by the KOG database in horsegram (MUN_r1.1_cds), *P. vulgaris* (Pvulgaris_218_v1.0), *V. angularis* (Vangularis_v1.a1), *L. japonicus* (Lj_r3.0), and *A. thaliana* (Araport11). (**A**) Numbers and ratios of genes annotated by the KOG database. (**B**) Ratio of the classified KOG categories in hit genes.

## DECLARATIONS
## LIST OF ABBREVIATIONS

BUSCO: benchmarking universal single-copy ortholog; CDS: coding sequence; CNV: copy number variation; CSK-HPAU: CSK Himachal Pradesh Agricultural University; GO: Gene Ontology; Kbp: kilobase pairs; KOG: euKaryotic clusters of Orthologous Groups; Mbp: megabase pairs; MP: mate-pair; NCBI: National Center for Biotechnology Information; NJ: neighbor-joining; PE: paired-end; SLR: synthetic long read; SNP: single nucleotide polymorphism; SSR: simple sequence repeat; TAS: target amplicon sequencing; TE: transposable element.

## ETHICAL APPROVAL

Not applicable.

## CONSENT FOR PUBLICATION

Not applicable.

## COMPETING INTERESTS

The authors declare that they have no competing interests.



**Figure 10.** Comparative and phylogenetic analyses with other legume species. (**A**) Graphical view of syntenic relationships between horsegram and *P. vulgaris* (left), *V. anagularis* (middle), and *L. japonicus* (right). Pink and blue dots show homologous sequences of MUN_r1.11 with forward and reverse directions against the reference sequences. (**B**) Distribution of Ks values of orthologous gene pairs in horsegram (Mu) and the four plant species: *P. vulgaris* (Pv), *V. anagularis* (Va), *L. japonicus* (Lj), and *A. thaliana* (At). **C:** Phylogenetic tree of 4,154 common single-copy genes of the six legume species: *P. vulgaris*, *V. anagularis*, *G. max*, *L. japonicus*, *M. truncatula*, and *A. thaliana*.

## FUNDING

This work was supported by the Bilateral Joint Research Projects from the Japan Society for the Promotion of Science and the Department of Science and Technology of the Government of India, and by funds from the Kazusa DNA Research Institute Foundation.

## AUTHOR CONTRIBUTIONS

The project was designed by S.I., R.C., and T.S. K.S. contributed to the genome sequencing, linkage map, and pseudomolecule construction. R.C. contributed to the creation of the mapping population. H.H. contributed to the genome assembly and gene prediction, annotation, and phylogenetic analysis. S.N. contributed to the transcript sequencing. H.N. contributed to the pseudomolecule construction. S.I. contributed to the entire process of data analysis.

## ACKNOWLEDGEMENTS

We thank A. Watanabe, S. Nakayama, Y. Kishida, M. Kohara, H. Tsuruoka, C. Minami, and S. Sasamoto (KDRI) for their technical assistance. We also thank the National Bureau of Plant

Genetic Resources (NBPGR), New Delhi, for providing the germplasm lines for diversity analysis.

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
