## [Reviewer Report]

Reviewer name and names of any other individual's who aided in reviewer Tianzuo WangDo you understand and agree to our policy of having open and named reviews, and having your review included with the published papers. (If no, please inform the editor that you cannot review this manuscript.)YesIs the language of sufficient quality?YesPlease add additional comments on language quality to clarify if needed
It can be improved better.Are all data available and do they match the descriptions in the paper? YesAdditional CommentsAre the data and metadata consistent with relevant minimum information or reporting standards? See GigaDB checklists for examples <a href="http://gigadb.org/site/guide" target="_blank">http://gigadb.org/site/guide</a>YesAdditional CommentsIs the data acquisition clear, complete and methodologically sound?YesAdditional CommentsIs there sufficient detail in the methods and data-processing steps to allow reproduction?YesAdditional CommentsIs there sufficient data validation and statistical analyses of data quality? YesAdditional CommentsIs the validation suitable for this type of data?YesAdditional CommentsIs there sufficient information for others to reuse this dataset or integrate it with other data?YesAdditional CommentsAny Additional Overall Comments to the AuthorShirasawa et al. reported a Chromosome-scale draft genome sequence of horsegram, and performed the analysis of comparative genomics.
1.If Pacbio data was used, the quality of genome can be improved much.
2.Only genomes of P. vulgaris, V. angularis and L. japonicus in the legume were used for phylogenetic analysis. Soybean and Medicago, as the model legume plants, should be added at last.
3.The section of Whole genome structure in horsegram should be introduced before Diversity analysis in genetic resources, Genes related to drought tolerance, and Transcript sequencing, gene prediction and annotation. Because genome information is the foundation of other analysis.RecommendationMajor Revision

---

## [Reviewer Report]

Reviewer name and names of any other individual's who aided in reviewer Penghao WangDo you understand and agree to our policy of having open and named reviews, and having your review included with the published papers. (If no, please inform the editor that you cannot review this manuscript.)YesIs the language of sufficient quality?YesPlease add additional comments on language quality to clarify if needed
Are all data available and do they match the descriptions in the paper? YesAdditional CommentsAre the data and metadata consistent with relevant minimum information or reporting standards? See GigaDB checklists for examples <a href="http://gigadb.org/site/guide" target="_blank">http://gigadb.org/site/guide</a>YesAdditional CommentsIs the data acquisition clear, complete and methodologically sound?YesAdditional CommentsIs there sufficient detail in the methods and data-processing steps to allow reproduction?YesAdditional CommentsIs there sufficient data validation and statistical analyses of data quality? YesAdditional CommentsIs the validation suitable for this type of data?YesAdditional CommentsIs there sufficient information for others to reuse this dataset or integrate it with other data?YesAdditional CommentsAny Additional Overall Comments to the AuthorAuthors presented a paper on describing a new pseudo-chromosome draft genome sequences of a legume plant horsegram and some bioinformatics analyses based on the data. The presented assembly is of good quality and the bioinformatics analysis performed is sound. The resources made available by the study should prove valuable to researchers working on the plant and legume community on a whole. The paper is generally well written and I personally found out the paper is quite easy to follow. Few grammatic errors can be found. 
The bioinformatics methodology that has been utilised in the study is sound and the software used fit the goals of the study. However, authors need to present more details on some analysis components, e.g. the parameter set used for the software, the version of the software, the OS, etc, so that the analysis can be better reproduced. For example, in Methods section, line 76 the Jellyfish program was used to estimate the genome size, the parameter, version, OS of running the software were not mentioned. Line 78 SOAPdenovo2 apart from Kmer the most important parameter, what about the rest? SSPACE 2.0 was used for scaffolding, the insert sizes? Platanus, MaSuRCA, TruSPAdes, RepeatMasker, augustus, all these software involve a number of parameters, and the details on how they were used need to be provided. Because the results can be sharply different with different parameters. Some figures appear to be created by using some tools, and these tools need to be acknowledged and referenced. For example, is Circus used to generate the circular plot in Fig 5? In addition, I could not find captions for all the main figures. 
RecommendationMinor Revision